# Research Progress in Modifications, Bioactivities, and Applications of Medicine and Food Homologous Plant Starch

**DOI:** 10.3390/foods13040558

**Published:** 2024-02-12

**Authors:** Kai Chen, Pinghui Wei, Meiqi Jia, Lihao Wang, Zihan Li, Zhongwei Zhang, Yuhuan Liu, Lin Shi

**Affiliations:** 1Shangrao Innovation Institute of Agricultural Technology, College of Life Science, Shangrao Normal University, Shangrao 334001, China; ccdatabank@163.com (K.C.); 17507086686@163.com (P.W.); 2College of Food Science, Shenyang Agricultural University, Shenyang 110866, China; 17856267792@163.com (M.J.); 2021240293@stu.syau.edu.cn (L.W.); 3State Key Laboratory of Food Science and Resources, Engineering Research Center for Biomass Conversion, Ministry of Education, College of Food Science and Technology, Nanchang University, Nanchang 330047, China; 352313320026@email.ncu.edu.cn (Z.L.); redmaple9966@163.com (Z.Z.)

**Keywords:** starch, edible medicinal plants, modification methods, functionality

## Abstract

Starchy foods are an essential part of people’s daily diet. Starch is the primary substance used by plants to store carbohydrates, and it is the primary source of energy for humans and animals. In China, a variety of plants, including edible medicinal plants, such as Pueraria root, yam tuber and coix seed, are rich in starch. However, limited by their inherent properties, kudzu starch and other starches are not suitable for the modern food industry. Natural starch is frequently altered by physical, chemical, or biological means to give it superior qualities to natural starch as it frequently cannot satisfy the demands of industrial manufacturing. Therefore, the deep processing market of modified starch and its products has a great potential. This paper reviews the modification methods which can provide excellent functional, rheological, and processing characteristics for these starches that can be used to improve the physical and chemical properties, texture properties, and edible qualities. This will provide a comprehensive reference for the modification and application of starch from medicinal and edible plants.

## 1. Introduction

The essence of edible medicinal plants is based on the concept of “medicine and food homology” guidance. These herbs, which are used in both Chinese traditional medicine and food therapy, can be both edible and medicinal [1]. They are often used in the medical care field in the form of dietary therapy, dietary supplements and medicinal diets [2].

Starch is the main reserve polysaccharide of plants which is divided into amylose and amylopectin. It is composed of *D*-glucose bonded by *α*-1,4 and *α*-1, 6-glycosidic bonds. Its shape and size vary depending on its biological source, and different types of natural starch have varying amylopectin contents, crystal structures, and gelatinization properties [3,4]. With the advancement of science and technology, new processes and equipment are widely used, and with the characteristics of natural starch being used at the same time, the scope of application is becoming narrower and narrower. 

In this review, our primary aim is to comprehensively explore the various modifications applied to starch of edible medicinal plants and their implications in the food industry. We will discuss the role of modified starches in modern industrial production and their contribution to expanding the application range of starch-based products. Specifically, we will focus on commonly used modified starches for food processing, including cold water-soluble starch [5], dextrin [6], acid-modified starch [7], crosslinked starch [8], hydroxypropyl starch [9], carboxymethyl starch [10], and starch phosphate [11]. Furthermore, we will delve into the specific applications and advantages of phosphate ester starch, hydroxypropyl distarch phosphate, and acetylated distarch phosphate, which are among the most commonly used modified starches in the food industry [12,13,14]. Our review aims to provide valuable insights into the role of modified starches in enhancing food quality, texture, and functionality, thus contributing to a broader understanding of their significance in the field of food science and technology.

## 2. Common Starch-Rich Medicinal and Edible Homologous Plants and Their Starch Characteristics

Starch is the primary ingredient of Pueraria root, yam tuber, coix seed, lotus seed and other edible medical substances included in the Chinese edible medicinal substances catalog. Their biological activity and active constituents are shown in Table 1.

### 2.1. Pueraria Starch (kudzu starch)

*Puerariae lobatae* Radix, a traditional Chinese medicine for both medicine and food, has long been used in Asian countries such as China, Japan and South Korea as a muscle relaxant, antipyretic, antidysentery, and for the treatment of hypertension [24]. It is generally believed that the main bioactive components of *P. lobata* are isoflavone small-molecule compounds, including puerarin, daidzein, soybean saponin, and their derivatives, which have multiple health care effects such as antioxidant, anti-inflammatory, antitumor, and hypoglycemic effects [25]. In addition to isoflavones, starch is another important component in *P. lobate*, and its starch content can reach 51.60% ((*w*/*w*), dry basis), meaning it has the potential to be used as a natural starch source. Kudzu starch is popular as a functional food and is usually eaten as a health drink after being prepared with hot water.

Although kudzu starch has rich edible value and application potential, studies have pointed out that natural kudzu starch has poor performance in terms of solubility and thermal stability [26]. This results from the fact that kudzu starch granules are easy to agglomerate after being directly added to hot or boiling water, forming a heterogeneous paste, which seriously limits its utilization and promotion in production and processing. The functional properties of starch are inseparable from the structure, and the amorphous region of starch granules is mainly composed of amylose, which plays an important role in the properties of starch products [27]. The amylose content of kudzu starch is 20–22% [24], which is high, causing causes kudzu starch to not swell in boiling water and making it difficult to be gelatinized. In order to meet the needs of industrial production for the intensive processing of kudzu starch, it is necessary to improve the properties of kudzu starch.

### 2.2. Lotus Seed Starch

Lotus seed is an important special economic resource in China, and is the fruits or seeds of *Nelumbo nubifera* Gaertn, which has been listed by the Ministry of Health of China as 1 of 87 medicinal and edible foods. Lotus seed has the functions of being antioxidative, hypolipidemic, and hypoglycemic and regulating gastrointestinal function [28]. Starch is the main component of lotus seed; starch content in Lotus seed can reach over 60% on a dry basis, of which the amylose content is as high as 42%, which belongs to the specific starch with high amylose content, and the granular crystal shape is a C-type structure [29]. The physicochemical changes in the crystal morphology and structure of starch granules will directly affect the quality of the product, such as structural properties, sensory properties, and nutritional properties [30]. With the maturity of lotus seeds, the starch content in a single cell increases, and the surface of starch grains changes from smooth to rough [31]. The diameter of the granules is between 3 and 30 μm, with an average of ~12 μm [32]. Due to the high amylose content of lotus seed starch aqueous solution, after high temperature treatment, it is easy to encounter a series of problems such as low transparency, high gel hardness, poor dispersion stability, and thick texture [33].

Therefore, in the application of lotus seed starch, it is necessary to use modification methods to improve its processing characteristics. Many previous studies have demonstrated its physicochemical properties and expanded the application of modifications, thereby improving and expanding the application of lotus seed starch, such as dry heat treatment [34], microwave heat treatment [35], retrogradation [36], heat and humidity treatment [37], and combination methods [38].

### 2.3. Yam Tuber Starch

Yams are the fourth most important type of potato in the world, after cassava, potato and sweet potato, which is rich in starch (66.21~68.88%, dry basis) [39]. Its particle size distribution ranges from 10 μm to 40 μm, and the content of directly connected starch is higher. Yam starch also has the characteristics of easy gelatinization, strong water absorption and expansion, stable viscosity of starch paste at high temperature and high gel strength [40].

More importantly, yam is used as a traditional Chinese medicine with a long history of medicinal use, in addition to its food value. Yam has antiaging, hypoglycemic, hypo-lipidemic, and immune-enhancing medicinal effects due to its functional active ingredients (polyphenols, allantoin, polysaccharides, saponins, etc.) [41,42]. The reason for the wide variation in the medicinal efficacy of different varieties of yam is attributed to the differences in the content of functional constituents in yam [43]. Yam polysaccharides, which are complexes of homopolysaccharides, heteropolysaccharides, and glycoproteins, make up the majority of the active components of yam, and have a broad range of medicinal effects. The monosaccharides currently found in polysaccharides are glucose, altrose, galactose, mannose, rhamnose, fucose, fructose, and xylose, among which the polysaccharides extracted from Dioscorea root are mainly composed of glucose, mannose, galactose, and glucuronic acid at a ratio of 1.2: 0.5: 0.3: 0.3 [41]. The composition and content of yam polysaccharides are influenced by a variety of factors (source, species, monosaccharide composition, molecular weight, length of branched chains, active groups, glycosidic linkages, and high-level structure), resulting in different medicinal effects of different yam varieties.

Starch, as the main polysaccharide in yam, has a promising application in yam medicinal use and health food development. Yam starch has a higher linear amylose content of about 30% compared to that found in maize (25%) and cassava (19%) [44], indicating that yam is a high-quality raw material for producing amylopectin and resistant starch. However, multiple factors, namely, unfavorable protection of yam varieties, insufficient research on pharmacological activities, browning phenomenon, and safety, are the main problems hindering the development and utilization of yam. Therefore, it is worthwhile to conduct further research exploring the functional active ingredients of different yam varieties and their mechanisms of action, elucidating the dose–effect relationship between different varieties of yam, and developing low-cost color protection agents with high color protection efficiency, which is of great significance to increase the yield of yam, accelerate its further processing and promote the development of medicinal and food-stuffs.

### 2.4. Coix Seed Starch

Coix seed, as a traditional Chinese food and medicine health food, has the effect of clearing heat and draining pus, is anti-inflammatory and analgesic, invigorates the spleen and dispels dampness, is a diuretic, enhances immunity, and reduces the risk of cancer because of the presence of coix seed oil, esters of coix seed, polysaccharides, proteins, and polyphenols [45,46]. Starch is the main component of coix seed, constituting about 70% of dry weight, of which the normal linear starch (five genotypes) ranges from 15.9 to 26.4%, while the waxy genotypes (five genotypes) contain less linear starch, whose thermal and pasting properties are similar to corn starch [47,48]. Previous studies on the pasting properties of coix starch showed that the values of pasting Tp (peak pasting temperature) and enthalpy change (ΔH) of the eight varieties were relatively close to each other, with values of 71–76 °C and 7–11 J/g, respectively [47]. Compared with maize and potato, which are the main sources of starch, there have been few studies on the structure and physicochemical properties of coix starch.

Compared with commercial starch, coix starch has the advantages of uniform particle shape, small particle size (3–14 μm), slow coagulation, and low digestibility, but it is difficult to be pasted and its highly hydrophilic hydroxyl group makes it susceptible to water erosion, which is a major limitation of its application [49]. Moreover, coix seed is difficult to cook due to its dense structure and hard texture, which also hinders its consumption. At present, the barley industry is still in the primary processing stage; the processing stage produces sugars, proteins, bran, chaff, roots, stems and leaves, and other wastes which must be further developed and utilization. Therefore, an in-depth understanding of the physicochemical properties of coix starch, the establishment of efficient means to improve the stability of starch granules and the development of a simple and convenient pre-treatment technology for coix seed are of great significance to improve the social and economic value of coix seed products.

## 3. Modification Methods and Physicochemical Properties of Modified Starch in Edible Medicinal Plants

Natural starch has limited applications due to its insolubility in cold water, lack of paste stability, poor aging and film-forming qualities, and low shear resistance. To improve these deficiencies, physical, chemical, and biological methods can be employed to alter the original properties of starch (such as solubility, color, fluidity, etc.) and obtain the desired characteristics [50]. Starch denaturation can be divided into three categories based on its reaction mechanism: starch degradation products, crosslinked starch and starch derivatives.

### 3.1. Physical Modification

Physical modification is a widely used technique to alter the properties of starch, such as its physical and chemical characteristics, crystal structure, and digestibility [51]. This method involves the application of mechanical force, the blending of non-starch polysaccharides and starch, heat treatment, and other techniques. It is a low-pollution, safe, and easy-to-operate process. The most common physical modification techniques are pre-gelatinization, mechanical grinding, wet heat treatment, dry heat treatment, extrusion, ultra-fine grinding, irradiation, and ultrasonic technologies.

#### 3.1.1. Pre-Gelatinization

Pre-gelatinized starch is frequently dried using spray drying, drum drying, and extrusion techniques. Its qualities are comparable to those of a hydrophilic colloid and it dissolves easily. It also does not require cooking before consumption [52].

Shi [53] used the two-drum drying method for hot paste, resulting in an instantaneous natural kudzu powder with 73.2% starch and 26.8% white sugar that can be produced by the drum at a steam pressure of 4 kg, providing high quality and flexibility. The pre-gelatinization of yam tuber powder was generally achieved by steaming diced cubes in an autoclave at 68950 Nm^−2^ for 5 min, and at 98 ± 2 °C for 10–30 min. The amylopectin content in pre-gelatinized yam tubers decreases with longer cooking times [54].

Pre-gelatinized starch has good water solubility, absorption, and expansion, which can effectively improve the taste and texture of food.

#### 3.1.2. Heat Treatment

Studies have shown that baking at temperatures above 140 °C causes the *P. lobata* starch particles to cluster together and increase significantly in size and distribution. The consistency index of kudzu starch paste dramatically drops after baking at 150–160 °C [55].

Furthermore, Feng’s research has revealed that the original A-type of lentil starch crystals remain after heat-moisture treatment (HMT); the crystallinity increases, the particle morphology remains largely unchanged, and the temperature of gelatinization, solubility, water absorption rate, antiaging property, freeze-thaw stability, transmittance, viscosity, expansion force, and oil absorption rate decrease [56]. The effects of HMT on the structure and characteristics of the starches from big coix seed (BCS), translucent coix seed (TCS), and small coix seed (SCS) were investigated. HMT starches’ gelatinization temperatures rose but their thermal enthalpies decreased. HMT starches demonstrated superior flowability due to the altered rheological features, particularly for TCS starch. Moreover, HMT has been found to reduce the amount of slowly digested starch and increase the amount of resistant and quickly digested starch [48]. But hydrothermal treatment can raise the temperature at which yam starch gelatinizes, decrease pasting viscosity, and increase the amount of resistant and slowly digested starch by 4.40% and 3.73%, respectively [57]. These results reveal that HMT has different impacts on the structural and physicochemical properties of different starches.

#### 3.1.3. Microwave Treatment

Microwaves are electromagnetic waves with frequencies ranging from 300 MHz to 300 GHz, and are widely used in food production due to their efficiency, time-saving capacity, superior permeability, and other heating properties [58,59]. The A-type crystal structures of the coix seed starch can be preserved through a combination of heat, moisture and microwave treatment, resulting in an increase in amylase starch (17.48%), slowly digested starch (12.37%), and resistant starch (8.56%) compared to native starch [60]. However, this process also causes a decrease in solubility, swelling power, viscosity, attenuation value and setback value. The “attenuation value” measures a starch gel’s resistance to viscosity reduction during temperature cycling, while the “setback value” quantifies the increase in viscosity when starch retrogrades upon cooling.

Additionally, the amount of amylose dissolved in lotus seed starch after being microwave-treated decreased from 120 to 87.89 mg/g, and an increase in microwave intensity was observed to cause an apparent aggregation of starch particles, with a maximum average particle diameter of 26.37 μm—2.15 times larger than that of untreated starch. Furthermore, the starch paste’s elasticity modulus, rheology viscosity, and thixotropic properties also dropped [61].

#### 3.1.4. Ultrasound Treatment

Ultrasonic technology, an effective and environmentally friendly approach to modify starch, has a wide range of potential applications [62]. It can reduce the molecular weight of starch. Compared with other modification methods, ultrasonic treatment has the advantages of short action time, non-random degradation, simple operation, and no pollution, making it an important physical modification method [63].

According to Zhang et al., the solubility of KS improved by around 23% when exposed to ultrasonic conditions of a duration of 180 min, a temperature of 30 °C, power of 90% (324 W), and frequency of 40 kHz. Despite this, the freeze–thaw stability remained mostly unchanged from the original starch. An additional analysis was performed on the ultrasonic modified KS’s amylose concentration and particle size [64]. The solubility of starch is related to the amylose content. Some ultrasonic treatment conditions easily destroy the double-helix structure of amylose, resulting in a decrease in the reversibility of the double-helix and an increase in the solubility. At the same time, ultrasound destroys the integrity of starch particles, which easily leads to a decrease in water-holding capacity, and thus a decrease in freeze–thaw stability [65]. The selected ultrasonic conditions did not change the freeze–thaw stability of Pueraria starch, which may be due to the fact that the ultrasonic conditions did not cause significant damage to the starch particles, so the freeze–thaw stability remained unchanged under the condition of increased solubility.

The internal crystal structure of starch can be destroyed by ultrasonic treatment, which can also cause the surface of the starch sample to exhibit a porous structure. This can cause some starch molecules to break their glycosidic bonds, forming a specific number of molecular chains of the right length, and increase the dissolution of amylose molecules, which will encourage the formation of a double-helix structure [66].

#### 3.1.5. Ultra-High Pressure Treatment

Ultra-high pressure (UHP) processing, also known as non-thermal processing, involves sealing food raw ingredients in a UHP container after packing and processing them at a high pressure (100–1000 MPa) and certain temperature for a predetermined amount of time. This technique is used for sterilization, to alter material, create new organizational structure, and improve food quality by changing, denaturing, and eliminating bacteria and other microbes in food [67]. Following treatment at 600 MPa, lily starch’s shape was nearly entirely disrupted and its particle size increased, suggesting that the starch was almost entirely gelatinized. After being subjected to UHP treatment, the viscosities of lily starch at its peak, valley, breakdown, and final values were all reduced [68].

The surface of coix seed flour was destroyed by the UHP (600 MPa) modification, resulting in visible dents and a decreased size. The starch crystals were broken and the polarization cross was increasingly hazy [69]. Zhang et al. treated the starch of Taibai kudzu root under 300 MPa and 500 MPa. They discovered that while UHP treatment could significantly increase paste transparency, decrease freeze–thaw stability, and improve paste viscosity to pH value, it has little effect on the size and shape of starch particles and no discernible effect on gelatinization viscosity characteristics [70].

At temperatures of 55 °C, 60 °C and 75 °C, the starch solution treated with UHP significantly increased its swelling power and solubility; however, the opposite trend was observed at 85 °C and 95 °C. Following UHP treatment, lotus seed starch’s light transmittance dropped, and this drop persisted as storage time increased. The freeze–thaw stability and retrogradation of lotus seed starch were enhanced via UHP processing at 500 MPa for 10 to 50 min, as compared to native starch [71]. Lotus seed starch was combined with a water suspension by Guo et al. [72]. After subjecting lotus seed starch to UHP treatment at 500 MPa and 25 °C, it was observed that with an increasing holding period, the size of the amylose particles expanded. The complex of endogenous lipids in lotus seed starch improved the stability of the amylose by reducing its swelling.

Enhancing the coagulation and freeze–thaw stability of natural starch with UHP treatment is advantageous. This is related to the change in the internal structure of starch molecules and the decrease in its ability to bind with water caused by UHP treatment [73]. However, as treatment duration rises, starch particle damage increases as well, leading to an increase in coagulation and water precipitation rate.

### 3.2. Chemical Modification

Chemical modification is a chemical method which adds new functional groups to starch molecules or changes the size and particle properties of starch molecules. This method only affects a small percentage of the hydroxyl groups in the starch. And the modified starches can be classified into water-soluble (that is, acid–alkali treatment of starch), esterified, etherified, oxidized and crosslinked starch [74]. Their degree of substitution, although minor, modify the nature of the starch. Chemical modification reacts the starch with chemical reagents under certain conditions to make starch particles expand.

#### 3.2.1. Acid-Modified Starch

Acid hydrolysis is one of the oldest methods of starch modification. The mechanism of acid modification involves the attack of hydroxide ions on the oxygen atoms of the primary starch glycoside, which leads to cleavage and depolymerization [75].

The characteristics of Pueraria starch vary depending on the acid treatment (AT) applied. Taibai Pueraria starch was found to have the best transparency when 3% hydrochloric acid was added and hydrolyzed for 4 h at 50 °C; when 3% sulfuric acid was added and hydrolyzed for 4 h at a constant temperature of 50 °C, Taibai Pueraria starch exhibited the best hot paste stability. After three hours at 40 °C and 1% oxalic acid hydrolyzing, the starch had the best gelatinicity [76].

Response surface testing was used to enhance the ultrasonic acid preparation procedure of Ginkgo biloba resistant starch, with an ultrasonic power of 528 W, ultrasonic duration of 20 min, HCl concentration of 1.6%, and Ginkgo starch milk concentration of 30.6%. Subsequent confirmation revealed that the yield of *G. biloba* resistant starch under these conditions was 26.45 ± 0.06% [77].

Previous studies have revealed that lily starch undergoing AT resulted in a rougher surface, reduced light from the polarization cross, a gradual increase in starch particle size and particle size distribution range with concentration, a B-type crystal structure, and a decrease in crystallinity. Lily modified starch showed a reduction in swelling degree and a considerable improvement in solubility. The temperature at which gelatinization occurs rose, the viscosity dramatically reduced, and the enthalpy value decreased. The interaction between acid and starch molecules breaks the chain of starch molecules and exposes more hydrogen bonds to interact with water, and the interaction between starch molecules becomes tighter, the surface of starch particles is damaged, and the aggregation between starch particles increases the size of starch particles [78].

#### 3.2.2. Crosslinked Starch

Crosslinking is a common method for starch modification, using a variety of crosslinking agents such as sodium trimetaphosphate (STMP), sodium tripolyphosphate (STPP), phosphoryl chloride (POCl_3_), and epichloro hydrin (ECH) [79]. Compared with the original starch, crosslinked starch has the characteristics of higher gelatinization temperature, better freeze–thaw stability, and strong acid and alkali resistance [80].

Li [81] synthesized kudzu crosslinked starch utilizing STMP as the crosslinking agent. They varied the reaction temperature (35 °C, 45 °C, and 55 °C), the reaction duration (1 h, 2 h, and 3 h) and the quantity of crosslinking agent (5%, 10%, and 12%). Crosslinking was performed, and as a result, the transparency, solubility, and expansion rate of Pueraria starch were reduced, the freeze–thaw stability was strengthened, and the thermal stability was increased.

Liu [82] developed yam crosslinked starch with varying gelatinization temperatures, viscosity, and transparency by using STMP and sodium tripolyphosphate as crosslinking agents. As a result, transparent starch film with variable mechanical characteristics were produced. The most desirable features of modified starch were obtained when it was hydroxypropylated with 10–12% propylene oxide and crosslinked with a combination of 2% STMP and 5% STPP. This resulted in modified starch that showed no viscosity breakdown, good acid resistance, high freeze–thaw stability, and enhanced gel texture. Because of the loosening of the hydroxypropyl starch granule structure and the breaking of the hydrogen link between starch chains, starch reacts with phosphate more readily and expands into larger granules [83].

Bai’s thesis [84] investigated the preparation of starch sodium octenyl succinate using Ginkgo starch and lipase as a catalyst. Differential scanning calorimetry studies revealed that upon esterification, the starch gelatinized more readily. X-ray diffraction research revealed that following esterification, there is an increase in the amorphous area of starch, which might be due to subgelatinization. Kudzu starch was found to be advantageous in terms of acid and alkali resistance, freeze–thaw stability, and thermal viscosity stability following acetylation and crosslinking [85].

#### 3.2.3. Esterified Starch

Octenyl succinic anhydride (OSA) was used to esterify yam starch in aqueous slurry systems. X-ray diffraction, FTIR, scanning electron microscopy, and a fast viscosity analyzer were used to assess the physicochemical characteristics of the starch. The modified starch exhibited greater viscosities and a lower gelatinization temperature [86].

Zhang [68] prepared octenyl succinylated lily starch samples, and the starch system of pseudoplastic fluid showed a shear thinning phenomenon with the increase in shear rate. The hydrophobic groups were added in the modification process and the hydrophilic properties were retained, resulting in amphiphilic biopolymers.

Sun et al. [87] described the OSA esterification of lotus seed starch (LS). Hydrophobic groups of OSA partly replaced the hydroxyl groups of starch granules during the esterification step, producing LS with both hydrophilic and hydrophobic groups. The introduction of substituents reduced the bond strength between starch molecules, causing an increase in the swelling power and solubility of the modified starch.

#### 3.2.4. Carboxymethylated Starch

Wang et al. synthesized carboxymethyl kudzu root starch for the first time under different reaction conditions [88]. Using wide-angle X-ray diffraction, the crystallinity of Pueraria starch was discovered to reduce after undergoing carboxymethylation. Thermogravimetric and derivative thermogravimetric analyses demonstrated that carboxymethylation led to an improvement in thermal stability.

Zhao et al. investigated the rheological characteristics of carboxymethyl Pueraria starch and found that the viscosity of the solution increased with increasing concentration of starch [89]. However, when the concentration of NaCl was raised, the viscosity decreased and eventually became constant at concentrations higher than 1%. Furthermore, when the acid and base were strong, the viscosity of the solution was low, while it remained high and relatively unchanged when the pH was between 4.5 and 8.5. Carboxymethylated Pueraria starch can be completely dissolved in water at room temperature, and the water solubility is greatly increased. In addition, its water-holding capacity is strong, and its freeze–thaw stability is greatly improved.

#### 3.2.5. Methylated Starch

Methylated Pueraria starch was optimized by Zhou [90] to produce a semi-solid texture with a milky white hue. Its surface was found to be smoother and the particles were observed to cling together. When the degree of substitution was 0.41, or very close to the margin, the spreadability improved with the degree of substitution and reached 89.7%. The water-holding capacity of methylated Pueraria starch increased with the increase in substitution degree.

The fundamental characteristics of the methyl Pueraria starch demonstrated that the starch particles stuck together and lacked a full particle form in the surrounding region. Methyl starch has relatively little solubility in cold water and significantly more solubility after heating. Its freeze–thaw stability was found to be notably better than that of the original starch [91].

### 3.3. Enzymatic Modification

The principle of enzymatic modification involves hydrolyzing glycoside bonds in starch molecules using specific enzymes. These deposition enzymes can be categorized into several types based on the method of hydrolysis, such as *α*-deposition enzyme, *β*-amylase, glycosylase, debranching enzyme, and others [92,93].

The impact of pullulanase enzymatic hydrolysis on the physicochemical characteristics of granular kudzu starch was documented [94]. Research has indicated that the peak viscosity and breakdown value of kudzu starch decreased by 28.33% and 94.69%, respectively, during the enzymatic hydrolysis process. Additionally, the resistant starch concentration rose from 1.29% to 4.60%. Huang’s work [95] assessed the synergistic hydrolysis efficiency of kudzu starch with *α*-amylase and glucoamylase and built a model of enzymatic hydrolysis kinetics. Glucoamylase and *α*-amylase showed a strong synergistic relationship. A concentration of 20 U for *α*-amylase and a concentration of 36 U for glucoamylase was shown to be the ideal combination for both enzymes.

Zheng et al. identified the optimal conditions for the enzymatic breakdown and modification of adlay starch, which was a ratio of pullulanase to *β*-amylase of 1:1 [96]. Under these conditions, the initial gelatinization temperature was 60.1 °C, the degree of esterification of adlay starch liquefaction was 23.28%, and the content of adlay starch was 276.3%. Liu [97] increased the stability of the yam beverages by adding 20 U of *α*-amylase and 36 U of saccharification enzyme.

Enzymatic methods use specific enzymes to treat starch particles, which can achieve the purpose of fully swelling starch particles.

### 3.4. Complex Modification

In addition to their structural and functional modifications, these starches may also exert effects on blood lipid regulation through mechanisms related to their altered digestibility and interaction with gastrointestinal processes. Bao et al. [98] reported that the ultrasonic–autoclaving treatment purified *Semen coicis* resistant starch, enzyme–autoclaving treatment purified *S. coicis* resistant starch, and microwave–moisture treatment purified this resistant starch in comparison to high-amylose maize starch (HAMS). *Ginkgo biloba* starch was treated with enzymes and microwaved to create the resistant starch. Using the yield of the resistant starch as an evaluation metric, a single-factor experiment was used to determine the ideal preparation conditions for *G. biloba* resistant starch, including a 15% milk concentration of *G. biloba* starch, 750 W of microwave power for 90 s, the addition of 20 U/g of isoamylase, 50 °C, pH 5.8, and 3 h. The output of the resistant starch was 25.3% [99].

The structural, physical, and chemical characteristics of lotus seed starch nanoparticles (LS-SNPs), which were made via enzymatic hydrolysis (EH) with high-pressure homogenization (HPH), were examined by Wang et al. [100]. The molecular weight and particle size of LS-SNPs are significantly impacted by HPH treatments at various frequencies and pressures. The double-helix structure of LS-SNPs was changed by HPH treatment, as evidenced by the relative crystallinity of H-LS-SNP increasing and then decreasing when the homogenization pressure and frequency were raised. Native lotus seed flour (N-LSF) was altered in Sopawong’s study using pullulanase treatment (EP), heat-moisture treatment (HMT), and partial gelatinization (PG). Rapid digestible starch (RDS) was enhanced by PG, while resistant starch (RS) was reduced; amylose and RS levels were increased to 34.57–39.23% and 86.99–92.52% total starch, respectively, by HMT and EP [101].

Kudzu powder’s instant qualities are significantly impacted by agglomeration, which may be prevented during rehydration by using an extrusion process at low moisture levels (10%, 15%, and 20%). When the rehydration duration approached 60 s, the extrusion-based agglomeration rate of kudzu powder dropped from 42.41% to 0.84–1.46%. The powder puffed up as a result of the abrupt pressure dips that occurred during the extrusion process, which decreased the powder’s bulk density and allowed water to enter the powder more quickly [102].

The optimization of the enzymatic hydrolysis combined with the extrusion puffing process of yam powder was conducted using a single-factor test and orthogonal test, and the ideal process conditions were determined to be 180 °C for the barrel rear zone temperature, 16% moisture content, 24 kg feed per hour, 160 rpm screw speed, and 100 U/g of *α*-amylase. The water solubility index of the produced yam powder was 42.80%, which was 80.82% higher than that of extrusion puffing [103]. Muhammad et al. [104] investigated acid-hydrolyzed high-amylose *P. montana* starch (PMS) as an emulsifier and modified it with OSA. It demonstrated the potential application of pickering emulsions stabilized with OSA-modified PMS as effective food-grade delivery vehicles in functional foods. In Liang’s work, the impact of citrate esterification and surface gelatinization treatment on red rice bean starch was examined. The starch’s molecular structure changed as a result of the surface gelatinization that was caused by CaCl_2_, but the starch’s Maltese cross and growth ring structure remained unchanged. Furthermore, adding citrate esterification enhanced the original starch’s poor pasting ability, solubility, and swelling power in addition to further reducing its molecular weight [105]. Lastly, HMT at various moisture levels (15–35%) and AT with hydrochloric acid at five different concentrations (0.25–2.0 M) were used to modify the starch recovered from lily bulbs. The starch granules clustered as a result of HMT and AT, roughening their surface and increasing the particle size. In cooked samples, the starch containing 25% HMT had the highest resistant starch concentration (44.15%) [106].

## 4. Bioactivities

RS is found in food and medicinal-related plant starches as the main bioactive component (Figure 1). Although only a few studies have been conducted on the clinical use of RS in food and medication substances, as dietary fiber research has become more and more popular, an increasing number of studies have demonstrated the potential function of RS in the prevention and treatment of various illnesses [107,108] (Table 2).

### 4.1. Regulation of Intestinal Flora

RS can be used as a fermentation substrate to regulate the gut microbiota after partial or full fermentation in the large intestine, thereby promoting the growth of probiotic bacteria, such as *Lactobacillus*, *Bacteroides*, *Clostridium*, and *Bifidobacterium* in the intestine [109,110].

*P. Lobatae* Radix RS is a class of polysaccharides that may withstand intestinal enzymatic breakdown and eventually function as a substrate for gut microbial fermentation. It has the ability to alter the gut microbiota and both prevent and treat certain illnesses [111]. Through the cultivation of sugar-free MRS medium, Zeng et al. [112] investigated the impact of lotus seed RS (RS) on the proliferation of intestinal probiotics in vitro and discovered that RS3 mostly stimulated the growth of *Bifidobacterium* and *Lactobacillus*. It was also found to promote the growth of *Bifidobacterium* and *Enterococcus* and the secretion more bile saline hydrolase and hydroxysteroid dehydrogenase in the intestine [113]. Li et al. then discovered that the rough-surfaced and flawless crystalized purple yam RS could both prevent and enhance Bifidobacterium growth from gastrointestinal conditions [114]. Coix seed RS could significantly increase the concentration of short-chain fatty acids in the intestine of mice, reducing the number of pathogenic or potentially pathogenic flora such as *Clostridium*, *Porphyromonadaceae,* and *Rikenellaceae*, while also increasing the amount of beneficial intestinal microflora like *Bacteroidaceae*, *Butyrivibrio,* and *Prevention* [115].

### 4.2. Antidiabetic and Anti-Obesity Activity

RS appears in a range of dietary products to improve health and lower the glycemic index (GI), particularly in the management of diabetes. In Song’s research, kudzu RS dramatically lowered the blood levels of total cholesterol (TC), total triglycerides (TG), high-density lipoprotein (HDL), and low-density lipoprotein (LDL) in type 2 diabetes mellitus (T2DM) mice, along with the value of fasting blood glucose. After analyzing the insulin signaling sensitivity in liver tissue, it was shown that consuming various dosages of kudzu RS could increase the efficiency of insulin production by restoring the expression of IRS-1, p-PI3K, p-Akt, and Glut4 [116] (Figure 1).

In a study conducted by Wang et al. [117], the hypoglycemic effect of lotus seed RS (LSRS) was investigated in mice with T2DM. The results showed that LSRS improved lipid metabolism and markedly decreased blood glucose levels by 16.0–33.6%, as well as restoring serum insulin levels by 25.0–39.0%.

Yang et al. [118] prepared instant rice bean powder with low starch digestibility and low estimated glycemic index by microwave cooking. In complexation with ferulic acid (FA) and quercetin (QR), Maibam et al. [119] examined the physicochemical, structural, and in vitro starch digestibility of *Euryale ferox* kernel starch (EFKS). The complexation process caused a change in color, a decrease in swelling strength, and an increase in solubility. Additionally, the percentage of RS increased from 6.79% to 49.39% (10% FA) and 54.68% (10% QR), resulting in a significant decrease in the anticipated glycemic index.

According to Li’s research [120], the Ginkgo starch–lauric acid combination can successfully tackle obesity caused by high-fat diets. In mice, this complex has been seen to reduce body weight, TG, and TC, as well as improve the composition of the intestinal flora and hepatic fat formation.

### 4.3. Maintaining Healthy Levels of Blood Lipids

RS can down-regulate the expression of genes related to fat formation and enhance the expression of transporters related to lipids, thus playing a role in improving blood lipid [107]. *P. lobata* starch (PLS) can effectively reduce inflammation, dyslipidemia, and hepatic steatosis in mice with non-alcoholic fatty liver disease (NAFLD) caused by a high-fat, high-cholesterol diet. PLS was found to have a protective effect on NAFLD-associated gut dysbiosis, increasing the number of *Lactobacillus*, *Bifidobacterium*, and *Turicibacter* while decreasing *Desulfovibrio* [121].

Previous research on hyperlipidemic rats has revealed that Chinese yam starch can dramatically reduce levels of LDL cholesterol, triglycerides, and total cholesterol in the blood, with estimated decreases of 33.8%, 46.2%, and 27.5%, respectively [122].

### 4.4. Other Activities

In T2DM mice, kudzu RS can significantly lower the amount of serum inflammatory factors. Following a 4-week course of therapy with several dosages of kudzu RS, MCP-1 and TNF-α levels in T2DM mice were markedly reduced and showed a specific quantitative impact connection [86]. Research has demonstrated that when mice are given the right amounts of lotus seed resistant starch, organic acids like butyric, lactic, and succinic acid, along with short-chain fatty acids, in the gut help the body absorb minerals like calcium, magnesium, and iron by lowering the pH of the gut, making minerals more soluble, and stimulating the growth of intestinal wall cells [123].

Recent outcomes have demonstrated that the use of RS derived from lotus seeds reduced the symptoms of food allergies, including reduced body temperature and allergic diarrhea. Additionally, in ovalbumin (OVA)-sensitized mice, lotus seed RS reduced the rise in OVA-specific immunoglobulins and enhanced the Th1/Th2 imbalance [124].

**Table 2 foods-13-00558-t002:** Physiological functions of RS. (BW: body weight).

Plant Source	Physiological Functions	Models	Dosage	Reference
Pueraria root	Regulation of intestinal flora; anti-inflammatory; anti-diabetic; lowers blood lipids.	T2DM mice (male C57BL/6J mice, high-fat diet (HFD) feeding, and streptozotocin (STZ) injection). T2DM mice (male C57BL/6J mice, HFD feeding).	0.5, 2.5, and 5.0 g/kg BW/d, orally. 400 mg/kg BW/d, orally.	[115,121]
Lotus seed	Regulation of intestinal flora; regulation of intestinal flora; promotes the absorption of minerals; protects against food allergy.	*B. longum* and *L. delbrueckii* subsp., MRS plus 0.5 g/L L-cysteine. T2DM mice (male Kunming mice, HFD feeding, and STZ injection). BALB/c male mice, fed the basal diet. BALB/c female mice, sensitized twice by intraperitoneal injection of 100 µg OVO	20% and 30% LRS3, orally. 5%, 10%, and 15% RS, orally. 5%, 10%, and 15% LRS3, orally. 0.3 g/100 g BW/d LRS, orally.	[111,116,123,124]
Yam tuber	Regulation of intestinal flora; anti-obesity activity; lowers blood lipids.	Bifidobacteria in simulated upper digestive tract conditions. Hyperlipidemic rats (Kunming male Wistar rats, fed a high-fat diet).	20 g/L. 6.5 g/kg BW/d.	[113,122]
Coix seed	Regulation of intestinal flora; controls weight gain; develops immunity.	BALB/c male mice fed the basal diet.	0.2 mL/10 g BW/d.	[113]
Rice bean	Controls estimated glycemic index (eGI).	In vitro starch digestion.	-	[117]
Gordon Euryale seed	Lowers predicted glycemic index (pGI).	In vitro starch digestion.	-	[118]
Ginkgo seed	Improves obesity; improves hepatic fat accumulation; lowers blood lipids.	Male SD rats; HFD feeding.	The Ginkgo starch–lauric acid complex.	[119]

## 5. Application in Food Industry

Modified starch has good viscosity, transparency, and antiaging properties, and can be used as a thickener, stabilizer and food additive, and in other applications in food processing, including beverages, confectionery, frozen food, noodles, meat, condiments, and baked food (Figure 2).

The altered starch molecules include a variety of hydrophilic groups, including carboxylic and alcohol hydroxyl groups. To improve its freeze–thaw stability, functional groups such as phosphoric and acetic acids are added to the starch molecules, which decreases the intermolecular hydrogen bond and increases hydrophilicity [50]. This modification also increases storage stability and makes it difficult to regenerate at low temperatures while maintaining a high thickening capacity under high temperature and strong acid conditions. Additionally, it is effective in thickening liquids, puddings, jams, pie fillings, and spices, as well as preventing water from precipitating in meat foods, such as ham and other low-temperature refrigerated meals.

Carboxymethyl starch has surface activity and can be used as an emulsifier in ice cream. Octenyl succinate starch ester is hydrophilic and oleophobic, which can improve the stability of flavor and fragrance in beverages. Modified starch can be used to create food gels [125]. In the food industry, oxidized starch is used to produce sugar. Gummies have a high gel strength and pleasant taste. Milk candy produced from acid-modified starch is non-stick and elastic and can maintain its stability for a prolonged period. Starch that has undergone enzymatic hydrolysis, acid hydrolysis, crosslinking, and other treatments, combined with suitable physical methods, can produce a fatty, soft texture similar to fat. When added to food, it can not only satisfy people’s taste for fatty foods, but also reduce calorie intake (Figure 3).

RS has a good structural improvement effect, and has been extensively used in bread, biscuits, and other baked foods in recent years. In contrast to the addition of traditional dietary fiber, baked foods containing RS do not have a harsh taste and have an improved food texture.

## 6. Conclusions

Modified starch, due to its advantageous physical and chemical properties, is extensively used in the food industry for cooking, baking, and the development of functional food. It can enhance the quality of food and has beneficial physiological effects such as regulating intestinal flora, improving the intestinal barrier, regulating inflammatory factors, controlling blood sugar levels, and preventing fat accumulation, all of which can be beneficial for disease prevention and treatment, and ultimately improve overall health. However, at present, most of the studies on these substances remain in the animal experimental stage, with lack of clinical data support, and the specific mechanism of action in the human body has not been fully defined. Therefore, in the future, efforts should be made to develop and utilize modified starch in starch-containing medicine and food homologous plants.

## Figures and Tables

**Figure 1 foods-13-00558-f001:**
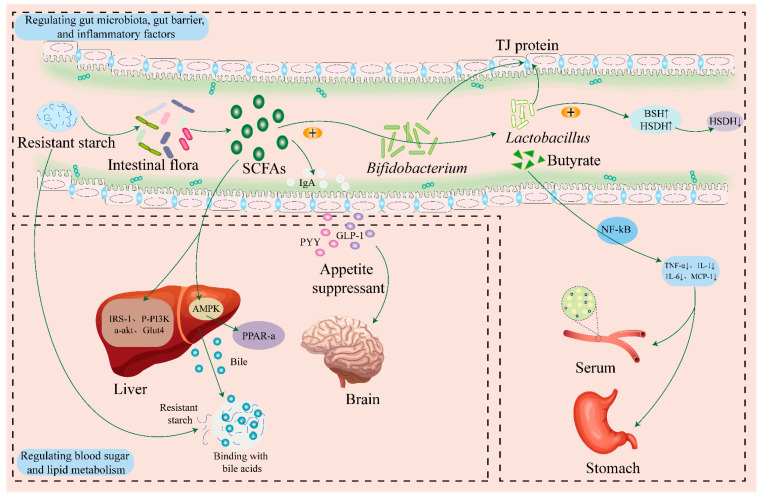
Effects of RS in medicine and food homologous plants on health.

**Figure 2 foods-13-00558-f002:**
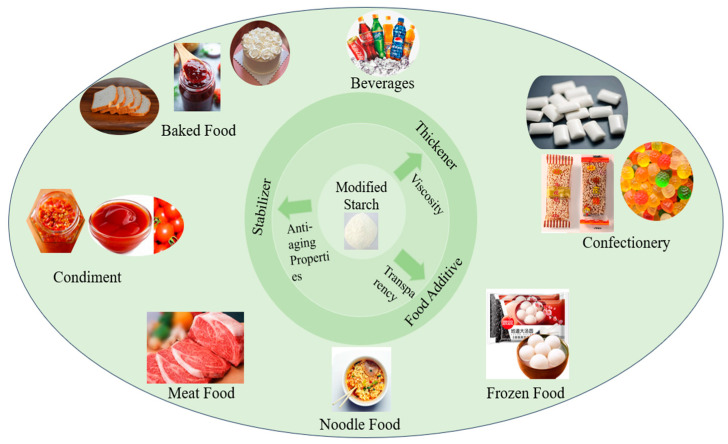
The application of modified starch in the food industry.

**Figure 3 foods-13-00558-f003:**
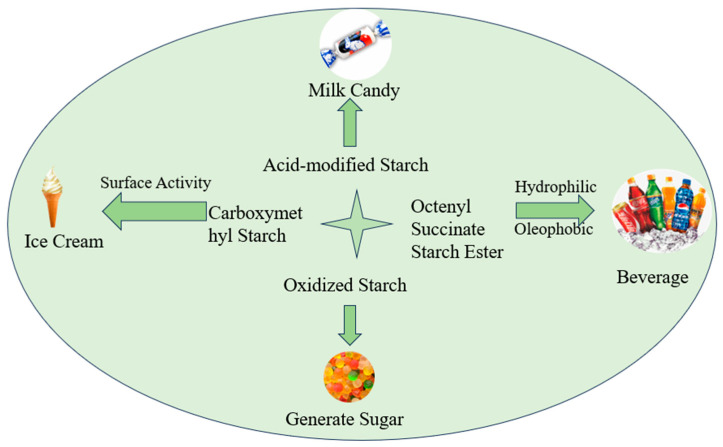
Application of several modified starches in the food industry.

**Table 1 foods-13-00558-t001:** The pharmacological action and active constituents of common starch-rich medicinal and edible homologous plants.

Plant Source	Pharmacological Action	Active Constituents	Reference
Pueraria root	Healing wounds and relieving fever, promoting eruption, promoting fluid production to quench thirst, ascending yang, and relieving diarrhea.	Flavonoids, triterpenes, coumarins, and organic acid	[15,16]
Lotus seed	Tonifying the spleen and stopping diarrhea, stopping band, tonifying the kidney and astringent essence, nourishing the heart, and calming nerves.	Starch, polysaccharides, and alkaloid	[17]
Yam tuber	Tonifying the spleen and stomach, generating fluid to nourish the lungs, and reinforcing the kidney to consolidate essence.	Steroidal saponins, polysaccharides, starch, flavonoids, phenolic glycosides, and fatty acids	[18,19]
Coix seed	Strengthening the spleen and tonifying the lungs, eliminating heat and dampness, removing pus and paralysis, and stopping diarrhea.	Esters, fatty acids, polysaccharides, phenolic acids, sterols, flavonoids, lactams, triterpenes, alkaloids, and adenosine	[20]
Rice bean	Eliminating heat and quenching thirst, decanting wine and detoxifying	Polyphenols, flavone, saponin, and polysaccharides	[21]
Gordon Euryale seed	Invigorating the spleen and stopping diarrhea, invigorating the kidney and reinforcing essence, removing dampness, and stopping belt.	Sterols, flavonoids, cyclic peptides, sesquineolignan, tocopherol, and cerebroside	[22]
Ginkgo seed	Reducing phlegm, eliminating poison, and treating diarrhea and frequent urination.	Flavonoids, terpene lactones, phenolic acid, polysaccharides, and organic acid	[23]

## Data Availability

Not applicable.

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
