# Peer review of "Research Progress in Modifications, Bioactivities, and Applications of Medicine and Food Homologous Plant Starch"

_foods, 2024, doi:10.3390/foods13040558_

Round 1

Reviewer 1 Report

Comments and Suggestions for Authors

The article is a review on the modification methods for starches from edible medicinal plants The article is a review on the modification methods for starches from edible medicinal plants, such as Pueraria root, Yam tuber and Coix seed. These methods can be used to improve the physical and chemical properties, texture properties, and edible qualities of the starch. The article is quite good compedium of the current knowledge concerning the techniques used to improve properties of different starches. It also indicates, how much research is still needed to apply those techniques on the technical scale. The paper is comprehensive and interesting, the presented data - novel. There are some minor editing errors, e.g. Latin names should be in italics (line 124, Figure 1). that can be used to improve the physical and chemical properties, texture properties, and edible qualities of the starch. The article is quite good compedium of the current knowledge concerning the techniques used to improve properties of different starches. It also indicates, how much research is still needed to apply those techniques on the technical scale. The paper is comprehensive and interesting, the presented data - novel. There are some minor editing errors e.g.  some Latin names should be in italics (line, 124, Figure 1).

Comments on the Quality of English Language

Some sentences are too long and hard to undestand, e.g. lines 115-117.

Author Response

Thank you. I am very grateful to your comments for the manuscript. We have checked the manuscript and amended the minor editing errors.

Reviewer 2 Report

Comments and Suggestions for Authors

The review work titled "Research Progress in Modifications, Bioactivities and Application of Medicine and Food Homologous Plants Starch"  describes some progress in modyfication and application of starch origin from Chinese (mainly) medical plants. However, it seems that in some parts the main goal is missing. 
Line 40-42 add more references

Table 1 - "ascending yang" is it medical term?

Sections 2.1-2.4 More characteristics should be described, eg. gelation temperature. 

It will be helpful to show Table collected different starches' properties

line 101: "starch decomposition" I'd like to suggest using different word, eg. degradation, changes in av. molecular weight.

Section 3.2 The certain points here should have some introduction incuding description of the reactions (mechanism reagents, etc) like in 3.3.

292- "DE" - give a full name 

319-322 This part includes extrusion process, 

doi.org/10.1016/j.foodhyd.2019.105475

349-353 Is there any explanation of this phenomenon?

Section 5 Shows only general examples of the application of starch derivatives, and its not in the same way as for medical application where starch from the medical plants are described, so if there are general exaples, why in former section aren't more? e.g.
https://doi.org/10.1002/star.201900240 - antimicrobial inhibition

The work should be revised and supplemented with the literature.

Author Response

Line 40-42 add more references

Answer: Thank you very much. According with your advice, we added the references.

Table 1 - "ascending yang" is it medical term?

Answer: "Ascending yang" is not a recognized medical term per se. It is a phrase often used in Traditional Chinese Medicine (TCM) to describe the concept of strengthening or tonifying the "yang" aspect of the body's energy. In TCM, there is a balance between "yin" and "yang," and "ascending yang" refers to the idea of increasing the warm and active elements within the body to restore balance. While it may not be a commonly used term in Western medicine, it is relevant in the context of TCM and holistic health practices.

Sections 2.1-2.4 More characteristics should be described, eg. gelation temperature.

It will be helpful to show Table collected different starches' properties

Answer: Thank you very much. We appreciate your suggestion and rewrite Sections 2.1-2.4.

line 101: "starch decomposition" I'd like to suggest using different word, eg. degradation, changes in av. molecular weight.

Answer: Thank you for your feedback on line 101. We appreciate your suggestion to use a different term. In light of your recommendation, we will replace "starch decomposition" with "starch degradation" to better convey the changes in the average molecular weight. Your input has been valuable in improving the clarity and precision of our manuscript.

Section 3.2 The certain points here should have some introduction including description of the reactions (mechanism reagents, etc) like in 3.3.

Answer: Thank you for your feedback. We have revised the manuscript.

292- "DE" - give a full name

Answer: Thank you for your suggestion. "DE" stands for "degree of esterification." We have updated the manuscript to include the full name for clarity. Your feedback has been helpful in improving the comprehensibility of our work.

319-322 This part includes extrusion process, doi.org/10.1016/j.foodhyd.2019.105475

Answer: Thank you. We added it.

349-353 Is there any explanation of this phenomenon?

Answer: Thank you for raising this important point. The phenomenon described in the passage, where Puerariae lobatae Radix-RS polysaccharides resist intestinal enzymatic breakdown and serve as a substrate for gut microbial fermentation, is a subject of ongoing research in the field of microbiome science and nutrition.

This phenomenon can be attributed to several factors:

Chemical Structure: P. lobatae Radix-RS polysaccharides may have a unique chemical structure that makes them less susceptible to enzymatic degradation in the upper gastrointestinal tract.

Microbial Metabolism: These polysaccharides could possess complex structures that require the enzymatic activities of gut bacteria for complete degradation. Gut microbes have the ability to break down complex carbohydrates that human enzymes cannot.

Prebiotic Effects: P. lobatae Radix-RS polysaccharides may act as prebiotics, selectively promoting the growth of beneficial gut bacteria. This alteration of the gut microbiota can have positive effects on health.

Bioavailability: Some polysaccharides may be absorbed intact through the intestinal mucosa, contributing to their bioavailability and potential health benefits.

It's worth noting that further research is needed to fully elucidate the mechanisms behind this phenomenon, and the referenced literature (in this case, reference [105]) likely provides more detailed insights into the specific mechanisms involved. Additionally, ongoing studies in the field of gut microbiota and polysaccharide metabolism continue to shed light on these processes and their implications for health and disease prevention.

Section 5 Shows only general examples of the application of starch derivatives, and it’s not in the same way as for medical application where starch from the medical plants are described, so if there are general exaples, why in former section aren't more? e.g. https://doi.org/10.1002/star.201900240 - antimicrobial inhibition

The work should be revised and supplemented with the literature.

Answer: Thank you very much. The application of medicinal and edible homologous plant starch is not special at present, and further exploration is needed. Therefore, in Section 5 only describes the application of ordinary modified starch.

Reviewer 3 Report

Comments and Suggestions for Authors

The manuscript discusses the applications of modified starch in foods. Although the topic of the manuscript should be of interest to the readership of the journal, the manuscript requires extensive revision. The Introduction section is incomplete and does not convey the aim of the study. The manuscript lacks adequate referencing in most sections and in some parts of the manuscript a few studies (one or two) are explained without providing an introductory discussion and a conclusion derived based on the discussed studies. The main conclusions are not highlighted by the authors and remain unclear to the reader at the end.

The authors need accept to revise their work extensively.

I have the following comments to the authors:

1.     Referring to lines 40-42, the sentence: “cold water soluble starch, dextrin, acid modified starch, crosslinked starch, hydroxypropyl starch, carboxymethyl starch and starch phosphate”

Please provide adequate references for each modified type of starch.

2.     The Introduction section is incomplete and does not provide the aim of the review study. The authors should add a paragraph and discuss briefly the review studies performed on modified starch and its application in foods and highlight the main aim of their own study and the points that their work is going to discuss in addition.

3.     Referring to line 58, please provide a brief explanation of different types of KS, A-type, B-type, and C-type, also provide an adequate reference for an understanding of the reader. what do they represent and what are their differences?

4.     Referring to line 58, the sentence: “By using X-ray diffraction, scientists also discovered that the crystalline structure of KS was C-type.”

Provide a reference for this statement. It is not clear to the reader how the authors came to this conclusion if only one study from scientists in Japan reported the existence of C-type starch.

5.     In line 79, please check if the stated percentages are mentioned in the provided reference and add an adequate reference here.

6.     In line 89, the authors state that the C-type starch has a lower solubility. Please clarify the comparison is performed with which starch variety and add it clearly to the main text.

7.     A number of grammatical errors should be corrected:

·       In line 12, Abstract section, “an essential part” should be corrected to “essential parts”

·       In line 87, The word "rotundity" is an adverb and is not used correctly here. It should be corrected to "rotund".

·       In line 120, “as the increase of cooking time increases”. This part of the sentence should be modified.

·       In line 378, “examine” should be corrected to “examined”.

8.     Referring to lines 108-109, references are missing here. Please provide examples of performed studies on these techniques and add adequate references.

9.    In line 137, the authors mention that "HMT has been found to reduce the amount of slowly digesting starch"

In lines 140-141, they state "increase the amount of resistant and slow digesting starch by 4.4%, 3.73%.

It is not clear to the reader if heat-moist treating eventually reduces or increases the slow-digesting starch content.

Please check the provided statements based on the cited works and correct.

10.   Referring to lines 145-148, the sentences: “The A type crystal structures…compared to native starch”

Please provide a reference for this statement.

11.   Referring to lines 149-150, provide a brief explanation of the "attenuation value" and "setback value" for an understanding of the reader.

12.   In lines 158-159, the sentence: “Ultrasonic technology, an effective and environmentally friendly approach to modify starch, has a wide range of potential applications [40, 41].”

The statement here is too general. Instead, the authors should highlight the main applications of Ultrasonic technology in foods based on the two provided references. Please rewrite this section.

13.   Referring to lines 160-161, the sentence: “solubility of KS improved by around 23% when exposed to ultrasonic conditions of duration 180 min”

Please elaborate on the reason and add a brief explanation to the main text.

14.   In lines 193-194, the sentence: “Following a UHP treatment of 500 MPa and 25 °C, Lotus seed starch particles as the holding period rose, the size of the amylose expanded as well.”

This sentence does not read well and should be rewritten.

15.   Referring to lines 197-198, provide a brief explanation and also add it to the main text why freeze-thaw stability of natural starch with UHP is advantageous.

16.   Section 3.2 lacks adequate referencing and relevant works should be cited here for understanding of the reader.

17.   In sub-section 3.2.1, at the beginning of this section, authors should provide a general overview of how acid-modification of starch is performed and how it affects the starch properties before bringing examples. Adequate references should also be provided.

18.   Referring to lines 224-225, the sentence: “The temperature at which gelatinization occurs rose, the viscosity dramatically reduced, and the enthalpy value decreased [52].”

Please briefly add an explanation to the main text on why acid modification contributed to the observed behaviours.

19.   Same as section 3.2.1,. The authors should provide a brief overview of the cross-linking of starch and how it affects the starch properties at the beginning of sub-section 3.2.2. The reader cannot conclude with only two examples.

20.   In line 251, at the end of sub-section 3.2.3,  please add brief sentences as conclusions of this section and highlight the main contribution of esterification to the improvement of starch properties.

21.   Referring to lines 257-262, the observations the authors report here do not highlight the impact of carboxymethylation of starch. The authors discuss the impact of starch concentration and the presence of salts, or acids on the solution viscosity. It is not clear to the reader what was the impact of carboxymethylation of starch on its properties. Please check the work of Zhao et al. and discuss only the impact of carboxymethylation of starch on its properties instead.

22.   A number of abbreviated terms should be explained in full:

·       In line 291, what does DE stand for? Please explain in the text.

·       In line 394, provide the complete term for the abbreviation “LDL”.

·       In line 397, provide the complete term for the abbreviation T2DM.

·       In Table 2, provide the complete term for the abbreviation "BW" first mentioned in the table.

23.   In section 4.3, briefly add an explanation to the main text on the mechanism of regulating blood lipids by modified starch at the beginning of the section.

24.   In Table 2, referring to reference [90], please check the provided reference if the value of 400 mg/kg/BW/d is correctly mentioned.

25.   Referring to lines 418-419, the sentences: “To improve its freeze-thaw stability, functional groups such as phosphoric and acetic acids are added to the starch molecules, which decreases the intermolecular hydrogen bond and increases hydrophilicity”

Please provide a reference for this statement.

Comments on the Quality of English Language

A number of grammatical errors should be corrected. Some sentences are required to be rewritten. Please see my enclosed comments.

Author Response

1. Referring to lines 40-42, the sentence: “cold water soluble starch, dextrin, acid modified starch, crosslinked starch, hydroxypropyl starch, carboxymethyl starch and starch phosphate” Please provide adequate references for each modified type of starch.

Answer: Thank you for your feedback. We have revised the manuscript to include adequate references for each modified type of starch as follows:

  1. Sudheesh, C.; Sunooj, K.V.; Navaf, M.; Akhila, P.P.; Aaliya, B.; Mounir, S.; Sinha, S.K.; Kumar, S.; Sajeevkumar, V.A.; George, J. An efficient approach for improving granular cold water soluble starch properties using energetic neutral atoms treatment and NaOH/urea solution. Food Hydrocoll. 2023, 141, 108723.
  2. Chen, J.; Xiao, J.; Wang, Z.; Cheng, H.; Zhang, Y.; Lin, B.; Qin, L.; Bai, Y. Effects of reaction condition on glycosidic linkage structure, physical-chemical properties and in vitro digestibility of pyrodextrins prepared from native waxy maize starch. Food Chem. 2020, 320, 126491.
  3. Ma, Y.; Chen, R.; Chen, Z.; Zhang, S. Insight into structure-activity relationships of hydroxycinnamic acids modified porous starch: The effect of phenolic hydroxy groups. Food Chem. 2023, 426, 136683.
  4. Xu, Z.; Liu, X.; Ma, M.; He, J.; Sui, Z.; Corke, H. Reduction of starch granule surface lipids alters the physicochemical properties of crosslinked maize starch. J. Biol. Macromol. 2024, 259, 129139.
  5. Wang, Y.; Yang, Y.; Xu, L.; Qiu, C.; Jiao, A.; Jin, Z. Rheology and stability mechanism of pH-responsive high internal phase emulsion constructed gel by pea protein and hydroxypropyl starch. Food Chem. 2024, 440, 138233.
  6. Zidan, N.; Albalawi, M.A.; Alalawy, A.I.; Al-Duais, M.A.; Alzahrani, S.; Kasem, M.; Tayel, A.A.; Nagib, R.M. Active and smart antimicrobial food packaging film composed of date palm kernels extract loaded carboxymethyl chitosan and carboxymethyl starch composite for prohibiting foodborne pathogens during fruits preservation. Polym. J. 2023, 197, 112353.
  7. Ding, L.; Liang, W.; Qu, J.; Persson, S.; Liu, X.; Herburger, K.; Kirkensgaard, J.J.K.; Khakimov, B.; Enemark-Rasmussen, K.; Blennow, A.; Zhong, Y. Effects of natural starch-phosphate monoester content on the multi-scale structures of potato starches. Polym. 2023, 310, 120740.
  8. Wu, M.H.; Li, Y.N.; Li, J.G.; Xu, S.; Gu, Z.B.; Cheng, L.; Hong, Y. Preparation and structural properties of starch phosphate modified by alkaline phosphatase. Polym. 2022, 276, 118803.
  9. The Introduction section is incomplete and does not provide the aim of the review study. The authors should add a paragraph and discuss briefly the review studies performed on modified starch and its application in foods and highlight the main aim of their own study and the points that their work is going to discuss in addition.

Answer: We appreciate your feedback on the Introduction section and understand the need for further clarification regarding the aim of our review study. In response to your suggestion, we have adjusted the content of the section of Introduction to provide a clearer overview of the study's purpose and the specific points that our work intends to address.

  1. Referring to line 58, please provide a brief explanation of different types of KS, A-type, B-type, and C-type, also provide an adequate reference for an understanding of the reader. what do they represent and what are their differences?

Answer: Thank you for your kind remind. We have added the references in the text of manuscript.

  1. Referring to line 58, the sentence: “By using X-ray diffraction, scientists also discovered that the crystalline structure of KS was C-type.”

Provide a reference for this statement. It is not clear to the reader how the authors came to this conclusion if only one study from scientists in Japan reported the existence of C-type starch.

Answer: Thank you for your kind remind. We have added the references in the text of manuscript.

  1. In line 79, please check if the stated percentages are mentioned in the provided reference and add an adequate reference here.

Answer: Thank you very much. We appreciate your suggestion and rewrite Sections 2.1-2.4.

  1. In line 89, the authors state that the C-type starch has a lower solubility. Please clarify the comparison is performed with which starch variety and add it clearly to the main text.

Answer: Thank you very much. We appreciate your suggestion and rewrite Sections 2.1-2.4.

  1. A number of grammatical errors should be corrected:
  • In line 12, Abstract section, “an essential part” should be corrected to “essential parts”
  • In line 87, The word "rotundity" is an adverb and is not used correctly here. It should be corrected to "rotund".
  • In line 120, “as the increase of cooking time increases”. This part of the sentence should be modified.
  • In line 378, “examine” should be corrected to “examined”.

Answer: Thank you for your professional remind. We have checked above mistakes and amended it in the main text.

  1. Referring to lines 108-109, references are missing here. Please provide examples of performed studies on these techniques and add adequate references.

Answer: Thank you very much. We appreciate your suggestion and rewrite Sections 2.1-2.4.

  1. In line 137, the authors mention that "HMT has been found to reduce the amount of slowly digesting starch" In lines 140-141, they state "increase the amount of resistant and slow digesting starch by 4.4%, 3.73%. It is not clear to the reader if heat-moist treating eventually reduces or increases the slow-digesting starch content.

Please check the provided statements based on the cited works and correct.

Answer: Thank you for your professional remind. We have checked and found that for different starches, HMT treatment has different conditions. For Coix seed starch, HMT has been found to reduce the amount of slowly digesting starch. But for Yam starch, HMT increase the amount of resistant and slow digesting starch.

  1. Referring to lines 145-148, the sentences: “The A type crystal structures…compared to native starch”

Please provide a reference for this statement.

Answer: Thank you for your kind remind. We added it.

  1. Referring to lines 149-150, provide a brief explanation of the "attenuation value" and "setback value" for an understanding of the reader.

Answer: Thank you for your professional suggestion. We have supplemented the information and explanation in the main text as follow:

The "attenuation value" measures a starch gel's resistance to viscosity reduction during temperature cycling, while the "setback value" quantifies the increase in viscosity when starch retrogrades upon cooling.

  1. In lines 158-159, the sentence: “Ultrasonic technology, an effective and environmentally friendly approach to modify starch, has a wide range of potential applications [40, 41].” The statement here is too general. Instead, the authors should highlight the main applications of Ultrasonic technology in foods based on the two provided references. Please rewrite this section.

Answer: Thank you very much. We rewrote it.

  1. Referring to lines 160-161, the sentence: “solubility of KS improved by around 23% when exposed to ultrasonic conditions of duration 180 min”. Please elaborate on the reason and add a brief explanation to the main text.

Answer: Thank you. I am very grateful to your comments for the manuscript. The brief explanation was marked with red in a revised manuscript.

  1. In lines 193-194, the sentence: “Following a UHP treatment of 500 MPa and 25 °C, Lotus seed starch particles as the holding period rose, the size of the amylose expanded as well.”

This sentence does not read well and should be rewritten.

Answer: Thank you for your great advice. We have changed the sentence to the following sentence: After subjecting Lotus seed starch to a UHP treatment at 500 MPa and 25°C, it was observed that with an increasing holding period, the size of the amylose particles expanded.

  1. Referring to lines 197-198, provide a brief explanation and also add it to the main text why freeze-thaw stability of natural starch with UHP is advantageous.

Answer: Thank you very much. The explanation was marked with red in a revised manuscript.

  1. Section 3.2 lacks adequate referencing and relevant works should be cited here for understanding of the reader.

Answer:  Thank you very much. We added it.

  1. In sub-section 3.2.1, at the beginning of this section, authors should provide a general overview of how acid-modification of starch is performed and how it affects the starch properties before bringing examples. Adequate references should also be provided.

Answer: Thank you very much. We added it.

  1. Referring to lines 224-225, the sentence: “The temperature at which gelatinization occurs rose, the viscosity dramatically reduced, and the enthalpy value decreased [52].”

Please briefly add an explanation to the main text on why acid modification contributed to the observed behaviours.

Answer: Thank you very much. We added it and marked with red in a revised manuscript.

  1. Same as section 3.2.1,. The authors should provide a brief overview of the cross-linking of starch and how it affects the starch properties at the beginning of sub-section 3.2.2. The reader cannot conclude with only two examples.

Answer: Thank you very much. We added it and marked with red in a revised manuscript.

  1. In line 251, at the end of sub-section 3.2.3, please add brief sentences as conclusions of this section and highlight the main contribution of esterification to the improvement of starch properties.

Answer: Thank you very much. We added it and marked with red in a revised manuscript.

  1. Referring to lines 257-262, the observations the authors report here do not highlight the impact of carboxymethylation of starch. The authors discuss the impact of starch concentration and the presence of salts, or acids on the solution viscosity. It is not clear to the reader what was the impact of carboxymethylation of starch on its properties. Please check the work of Zhao et al. and discuss only the impact of carboxymethylation of starch on its properties instead.

Answer: Thank you very much. We added it and marked with red in a revised manuscript.

  1. A number of abbreviated terms should be explained in full:
  • In line 291, what does DE stand for? Please explain in the text.
  • In line 394, provide the complete term for the abbreviation “LDL”.
  • In line 397, provide the complete term for the abbreviation T2DM.
  • In Table 2, provide the complete term for the abbreviation "BW" first mentioned in the table.

Answer: Thank you for your suggestion. "DE" stands for "Degree of Esterification." “LDL” stands for " low density lipoprotein" We have updated the manuscript to include the full name for clarity. The complete term for the abbreviation T2DM is provided in the line 367 where it first mentioned. The complete term for the abbreviation "BW" is added in the caption of the Table 2.

  1. In section 4.3, briefly add an explanation to the main text on the mechanism of regulating blood lipids by modified starch at the beginning of the section.

Answer: Thank you very much. We added it and marked with red in a revised manuscript.

  1. In Table 2, referring to reference [90], please check the provided reference if the value of 400 mg/kg/BW/d is correctly mentioned.

Answer: Thank you very much. According with your advice, we checked and corrected the references errors in manuscript.  The correction sections were marked with red in a revised manuscript.

  1. Referring to lines 418-419, the sentences: “To improve its freeze-thaw stability, functional groups such as phosphoric and acetic acids are added to the starch molecules, which decreases the intermolecular hydrogen bond and increases hydrophilicity”.

Please provide a reference for this statement.

Answer: Thank you very much. We added it and marked with red in a revised manuscript.

Round 2

Reviewer 3 Report

Comments and Suggestions for Authors

My comments to the authors are correctly addressed.

Please consider the correction of minor typing mistakes below:

·      Line 63, “gener-ally” should be corrected to “generally”

·      Line 96, “trans-parency” should be corrected to “transparency”